# Characterization of the Complete Mitochondrial Genome of *Drabescus ineffectus* and *Roxasellana stellata* (Hemiptera: Cicadellidae: Deltocephalinae: Drabescini) and Their Phylogenetic Implications

**DOI:** 10.3390/insects11080534

**Published:** 2020-08-14

**Authors:** Deliang Xu, Tinghao Yu, Yalin Zhang

**Affiliations:** Key Laboratory of Plant Protection Resources and Pest Management, Ministry of Education, Entomological Museum, College of Plant Protection, Northwest A&F University, Yangling, Shaanxi 712100, China; 18710305867@163.com (D.X.); 18404965218@163.com (T.Y.)

**Keywords:** Deltocephalinae, Drabescini, mitochondrial genome, phylogeny

## Abstract

**Simple Summary:**

Drabescini comprises over 225 species and 46 genera with a highly diverse tribe of Deltocephalinae. These species serve as vectors of numerous agricultural plant pathogens and also transmit plant viruses to host plants. Previous phylogenetic analyses in this tribe mainly focused on morphological characters and were restricted to several gene fragments. Furthermore, the taxonomic status and phylogenetic relationships of this tribe need to be further studied. Therefore, the mitogenome may provide additional molecular evidence to reconstruct the phylogeny of this group and further elucidate relationships among major lineages. In this study, we sequenced and analyzed two newly complete mitogenomes including *Drabescus ineffectus* and *Roxasellana stellata*. These two mitogenomes contain 13 protein-coding genes, two ribosomal RNA genes, 22 transfer RNA genes and the non-coding structure called A + T-control region. The Drabescini mitogenomes are highly conserved in base content and composition, genome size and order, protein-coding genes and codon usage, and secondary structure of tRNAs. Phylogenetic analyses using Bayesian inference and maximum likelihood methods indicated strong support for the monophyly of Drabescini. These results provide the comprehensive framework and valuable data toward the future resolution of phylogenetic relationships in this tribe.

**Abstract:**

To explore the mitogenome characteristics and shed light on the phylogenetic relationships and molecular evolution of Drabescini species, we sequenced and analyzed the complete mitochondrial genome of two species including *Drabescus ineffectus* and *Roxasellana stellata*. The complete mitogenomes of *D. ineffectus* and *R. stellata* are circular, closed and double-stranded molecules with a total length of 15744 bp and 15361 bp, respectively. These two newly sequenced mitogenomes contain the typical 37 genes. Most protein-coding genes (PCGs) began with the start codon ATN and terminated with the terminal codon TAA or TAG, with an exception of a special initiation codon of *ND5*, which started with TTG, and an incomplete stop codon T-- was found in the *Cytb*, *COX2*, *ND1* and *ND4*. All tRNAs could be folded into the canonical cloverleaf secondary structure except for the *trnS1*, which lacks the DHU arm and is replaced by a simple loop. The multiple tandem repeat units were found in A + T-control region. The sliding window, Ka/Ks and genetic distance analyses indicated that the *ATP8* presents a high variability and fast evolutionary rate compared to other PCGs. Phylogenetic analyses based on three different datasets (PCG123, PCG12R and AA) using both Bayesian inference (BI) and maximum likelihood (ML) methods showed strong support for the monophyly of Drabescini.

## 1. Introduction

Deltocephalinae is the largest subfamily of leafhoppers, presenting distinct diagnostic characteristics and including over 6600 described extant species and 39 tribes widely distributed in all zoogeographic regions [1]. They are now recognized as an ecologically and economically significant subfamily of leafhoppers. Drabescini, a highly diverse tribe of Deltocephalinae, contains approximately 225 species in 46 genera divided into two subtribes. Drabescini leafhoppers are often found on woody hosts and shrubs in Old World tropical or deciduous forests and have been collected at light [2,3]. These species feed on the sap of a variety of vascular plants via piercing-sucking mouthparts, serving as vectors of numerous agricultural plant pathogens and also transmitting plant viruses to host plants.

Previous studies in this tribe mostly concentrated on taxonomic descriptions and morphological characters of the nymphs [4,5,6,7]. Subsequently, morphological phylogenetic study was performed based on a species of this tribe [8]. Phylogenetic analyses of Drabescini among three genera including *Bhatia*, *Drabescus* and *Parabolopona* based on the morphological characters and molecular data (*28S* rDNA, histon *H3*) found it to form a monophyletic group with high branch support [1,9]. Recent phylogenomic analysis using the anchored hybrid enrichment method showed support for the monophyly of this tribe [10]. These phylogenetic analyses mainly depended on morphological characters and were restricted to several gene fragments. Moreover, the taxonomic status and phylogenetic relationships of this tribe need to be further studied based on more DNA data. Therefore, a new method examining the mitochondrial genome (mitogenome) may provide additional molecular evidence to reconstruct the phylogeny of this group and further elucidate relationships among major lineages.

The insect mitochondrial genome is typically a circular, closed and double-stranded DNA molecule with a total of 37 genes including 13 protein-coding genes (PCGs), two ribosomal RNA genes (rRNAs) and 22 transfer RNA genes (tRNAs) [11,12]. Additionally, it also has the non-coding structure called A + T-control region or A + T-rich region [11,12,13]. The mitochondrial genome provides important molecular material and is widely used in the study of insect phylogenetic relationships, evolution, population genetic structure and biogeography on account of its faster evolutionary rate, simple genetic structure, relatively stable composition, smaller length (ranging from 14 to 17 kb) and strict maternal inheritance [13,14,15]. However, only three species of Drabescini, *Athysanopsis* sp. (KX437726), *Drabescoides nuchalis* (NC_028154) and *Dryadomorpha* sp. (KX437736), representing three genera, have complete or partial mitogenome sequences available in GenBank [16,17].

In this study, we sequenced and annotated the complete mitochondrial genome of two additional species including *Drabescus ineffectus* (Walker, 1858) (GenBank accession no. MT527188) and *Roxasellana stellata* Zhang & Zhang, 1998 (GenBank accession no. MT527187). We reconstructed their phylogenetic relationships and confirmed their taxonomic status, and incorporating the previously published mitochondrial genome of Drabescini based on the concatenated nucleotide sequences of 13 protein-coding genes and two ribosomal RNA genes. Furthermore, we analyzed the complete mitochondrial structure of these two species, including genome size and nucleotide composition, codon usage, tRNA secondary structure, gene overlaps and intergenic spacers, evolutionary rate, and A + T-control region and made further comparisons with other Drabescini species. The purpose of this research is to test the monophyly of this tribe and analyze phylogenetic relationships among major lineages of this superfamily.

## 2. Materials and Methods 

### 2.1. Sample Collection and Genomic DNA Extraction

Specimens of *D. ineffectus* used in this study were collected from the Dachuan Town, Shiyan City, Hubei Province, 620 m, 3 July 2019, China, while *R. stellata* specimens were captured from the Diaoluoshan National Nature Reserve, Hainan Province, 15 July 2019, China. All fresh specimens were immediately preserved in 100% ethanol and stored at −20 °C in the laboratory. Identification of adult leafhoppers was based on external morphological characters and male genitalia. Total genomic DNA was extracted from abdomen tissues using the EasyPure Genomic DNA Kit (TransGen Biotech, Beijing, China) following the manufacturer’s protocol. Voucher specimens are deposited in the Entomological Museum of Northwest A&F University, Yangling, Shaanxi, China.

### 2.2. Mitogenome Sequencing, Assembly and Annotation

The whole mitochondrial genome sequences of these two species were generated using the next-generation sequencing (NGS) at the Illumina HiSeq™ Xten platform using the methodology of the PE150 (Biomarker Technologies, Beijing, China). The raw paired reads were retrieved and quality-trimmed selecting the mitochondrial genome of *Drabescoides nuchalis* (Jacobi, 1943) using reference sequences in the Geneious 8.1.3 (Biomatters, Auckland, New Zealand) with default parameters [18]. Then, the contig was assembled and annotated into the complete circular mitogenome in a similar way also using the Geneious 8.1.3 and *D. nuchalis* as a reference. The 13 PCGs were predicted by comparison with the homologous sequence of reference mitogenomes and finding the open reading frames (ORFs) based on the invertebrate mitochondrial genetic code Table 5. The locations of 22 tRNAs were identified by using the MITOS WebServer (http://mitos.bioinf.uni-leipzig.de/index.py) [19]. Their secondary structures were manually plotted with Adobe Illustrator CC2019 according to the MITOS predictions. The two ribosomal RNA genes (*rrnS* and *rrnL*) and the A + T-rich region were determined by the locations of adjacent genes (*trnL1* and *trnV*) and alignment with the homologous sequences of reference mitogenomes. Next, the mitogenomic circular maps were portrayed with CGView Server (http://stothard.afns.ualberta.ca/cgview_server/) [20].

### 2.3. Sequence Analyses

The nucleotide composition and skew, codon usage of PCGs and relative synonymous codon usage (RSCU) values of each PCG were calculated using PhyloSuite v1.2.1 [21], and tandem repeat units of the A + T-control region were analyzed with Tandem Repeats Finder online server (http://tandem.bu.edu/trf/trf.html) [22]. Strand asymmetry was calculated by using the formulas AT-skew = (A – T)/(A + T) and GC-skew = (G – C)/(G + C). A sliding window analysis concerning 200 bp and a step size of 20 bp was conducted by the DnaSP v6 [23] to estimate nucleotide diversity (Pi value) of 13 PCGs among four Drabescini mitogenomes. The ratio of the number of nonsynonymous substitutions per nonsynonymous site (Ka) to the number of synonymous substitutions per synonymous site (Ks) with regard to 13 PCGs of four species were also estimated with DnaSP v6. Genetic distances among the mitogenomes of the four species were calculated using MEGA X (https://www.megasoftware.net) under the Kimura 2-parameter model [24]. Complete mitogenome sequences of two species were deposited in GenBank and given the accession numbers MT527187 and MT527188 (Table 1).

### 2.4. Sequence Alignments and Phylogenetic Analyses

A total of 53 mitogenomes of Cicadomorpha insects were collected to analyze the phylogenetic relationships. Two newly sequenced specimens and 49 available mitogenomes of Membracoidea representing 13 subfamilies were selected as ingroups (Table 1). Two species, *Cryptotympana atrata* (Fabricius, 1775) (JQ910980) from Cicadoidea (cicadas) and *Cosmoscarta dorsimacula* (Walker, 1851) (NC_040115) from Cercopoidea (spittlebugs), were employed as outgroup taxa (Table 1). The nucleotide sequences of all 13 PCGs and two rRNA genes and amino acid sequences were used to elucidate the phylogenetic relationships of this tribe. All the available mitochondrial genomes were downloaded from GenBank for phylogenetic analyses (Table 1).

Complete and partial mitogenome genes were extracted using PhyloSuite v1.2.1. The nucleotide sequences of all PCGs of the 53 species were aligned in batches with the MAFFT v7.313 (https://mafft.cbrc.jp/alignment/server/) algorithm integrated into PhyloSuite v1.2.1, using the codon alignment mode and G-INS-i (accurate) strategy. The alignment of all rRNAs was conducted in the MAFFT version 7 online service with the G-INS-i strategy (https://mafft.cbrc.jp/alignment/server/) [46]. Then, gaps and ambiguous sites in the alignments were removed using Gblocks 0.91b [47] and alignments of individual genes were concatenated using PhyloSuite v1.2.1. Phylogenetic relationships based on three different datasets were generated: (1) a PCG123 matrix, including all three codon positions of 13 protein-coding genes with 10,749 nucleotides of 53 species; (2) a PCG12R matrix, including the first and second codon positions of 13 protein-coding genes plus two rRNAs with 8901 nucleotides of 52 species; (3) and an AA matrix, amino acid sequences of 13 protein-coding genes, with 3334 amino acids of 53 species. Because the *rrnL* and *rrnS* genes were missing in partial mitogenomes, *Dryadomorpha* sp. was excluded in the PCG12R analysis.

The optimal partitioning scheme and nucleotide substitution model for Bayesian inference (BI) and maximum likelihood (ML) phylogenetic analyses based on three different datasets were selected with PartitionFinder 2.1.1 incorporated into PhyloSuite v1.2.1, using the branch lengths linked, Bayesian information criterion (BIC) model and the greedy search algorithm [48] (Appendix A). The BI phylogenetic analysis was carried out using MrBayes 3.2.6 [49] with the following settings: two independent runs were run for four to thirty million generations with sampling every 1000 generations; four independent Markov Chain Monte Carlo (MCMC) chains were run, including three heated chains and a cold chain; a stationary phase was indicated after the average standard deviation of split frequencies < 0.01 and effective sample size (ESS) > 200; the initial 25% of samples were discarded as burn-in and the remaining samples were used to generate a consensus tree and estimate the posterior probabilities (PP). In addition, the ML phylogenetic analysis was conducted by IQ-TREE v.1.6.8 [50], using the ultrafast bootstrap (UFB) algorithm with 1000 replicates. Bootstrap support (BS) values were evaluated with 1000 replicates.

## 3. Results and Discussion

### 3.1. Mitogenome Organization and Nucleotide Composition

The complete mitochondrial genome of *D. ineffectus* (GenBank no. MT527188) and *R. stellata* (GenBank no. MT527187) are circular, closed and double-stranded molecules with a length of 15,744 bp and 15,361 bp, respectively (Figure 1). The genomes are of medium-sized sequence lengths compared to the other three Drabescini mitogenomes, ranging from 12,297 bp (*Dryadomorpha* sp., partial genome) to 15,309 bp (*D. nuchalis*) (Appendix A). The mutable size of mitogenomes among Drabescini species is mainly the variable length of the A + T-control region. These two newly-sequenced mitogenomes contain the typical 37 genes (13 PCGs, two rRNAs and 22 tRNAs) and the A + T-control region. The gene order is in accordance with original mitochondrial genome arrangements and other Drabescini mitogenomes. The majority strand (J-strand) generally encodes 23 genes including nine PCGs and 14 tRNAs. The remaining 14 genes are encoded on the minority strand (N-strand) and possess four PCGs, two rRNAs and eight tRNAs in these two mitogenomes (Appendix A). 

Nucleotide compositions of *D. ineffectus* are A = 41.7%, C = 13%, G = 9.9% and T = 35.4% and A = 41.4%, C = 14%, G = 9.9% and T = 34.6% in *R. stellata*. This exhibits a heavy AT nucleotide bias, with a high AT content for the entire sequence reaching 77.1% in *D. ineffectus* and 76% in *R. stellata* (Appendix A), respectively. This situation is also found in other Drabescini mitogenomes. The control region of *D. ineffectus* has the highest AT content with regard to the whole genome, PCGs and RNAs, but the PCGs have the lowest AT content. However, the rRNAs of *R. stellata* have the highest AT content, and the control region has the lowest AT content. Besides, the AT content in rRNAs is higher than PCGs and tRNAs in these two species. These two species showed a positive AT-skew (0.081, 0.089) and a negative GC-skew (−0.133, −0.17) in the whole genome, which also appears in other Drabescini species (Appendix A).

### 3.2. Protein-Coding Genes and Codon Usage

The total length of 13 PCGs with 10,956 bp for *D. ineffectus* and 10,932 bp for *R. stellata* accounts for 69.6% and 71.2% of their overall genomes, respectively (Appendix A). The size of 13 PCGs with the smallest gene was the *ATP8* and the largest gene was the *ND5* ranging from 153 bp to 1677 bp in this tribe. These two species show a negative AT-skew (−0.124, −0.099) and positive or negative GC-skew (0.009, −0.013) in PCGs. The AT content of the third codon (85.8%, 84.9%) was much higher than in the first (71.9%, 72.1%) and second codon positions (68.7%, 68.7%) in *D. ineffectus* and *R. stellata* (Appendix A). Across the 13 PCGs in these two species, only four PCGs (*ND1*, *ND4*, *ND4L* and *ND5*) were encoded on the N-strand, whereas the other nine PCGs (*COX1*, *COX2*, *COX3*, *ATP6*, *ATP8*, *ND2*, *ND3*, *ND6* and *Cytb*) were located on the J-strand (Figure 1 and Appendix A). In the Drabescini mitogenomes, all PCGs started with the putative codon ATN (ATA, ATT, ATG, ATC) except for the special initiation codon of *ND5*, which began with TTG in *Athysanopsis* sp., *Dryadomorpha* sp. and *R. stellate*; this has also been observed in other Deltocephalinae mitogenomes. Correspondingly, the PCGs ended with the putative terminal codon TAA or TAG, but an incomplete stop codon T-- was found in the *Cytb*, *COX2*, *ND1* and *ND4* among the five sequenced mitochondrial genomes. These incomplete termination codons may be converted into TAA by posttranscriptional polyadenylation during the mRNA maturation process [51], as has been reported in other leafhoppers. Therefore, the occurrence of termination codon TAA was more common than TAG and at least an incomplete stop codon T-- was present in all five mitogenomes.

The relative synonymous codon usage (RSCU) of five sequenced mitogenomes was calculated and is summarized in Figure 2. The results showed that the four most frequently utilized amino acids were Ile (AUU), Leu (UUA), Met (AUA) and Phe (UUU). Furthermore, they are merely composed of A or U, indicating the codon usage has a strong bias toward the nucleotides A and T and reflects the high AT content in the three codon positions of PCGs in Drabescini. This codon usage pattern of these two new mitogenomes highly resembles the pattern found in previously reported Cicadellidae species [32,34]. Additionally, the codon Ser1 (AGG) is absent in *R. stellata*.

### 3.3. Gene Overlaps and Intergenic Spacers

There are 11 gene overlaps in *D. ineffectus*, ranging in size from 1 to 8 bp and amounting to 43 bp, while *R. stellata* has eight gene overlaps ranging in the same size as the former and amounting to 32 bp (Appendix A). The longest overlap region in these two new mitogenomes was 8 bp between the *trnW*-*trnC* junction except for *D. ineffectus* which also had 8 bp between the *ND6-Cytb* junction. Correspondingly, the longest overlap also found in the other three known mitogenomes including *Athysanopsis* sp., *D*. *nuchalis* and *Dryadomorpha* sp. was 10 bp, 10 bp and 16 bp between *ND4*-*ND4L*, *trnW*-*trnC* and *ND4L-trnT* junctions, respectively. The complete mitogenomes of the four Drabescini species have one identical overlap region containing the *ATP8*-*ATP6* junction (7 bp) except for the partial mitogenome of *Dryadomorpha* sp.

As opposed to 12 intergenic spacers that occur in *D. ineffectus*, ranging in size from 1 to 20 bp and adding up to 68 bp, a total of 14 intergenic spacers were identified in *R. stellata* altogether, presenting 81 bp ranging in size from 1 to 17 bp. In these two new mitogenomes, the longest intergenic spacer was 20 bp in *D. ineffectus* and 17 bp in *R. stellata* between *trnY* and *COX1*, while in *Athysanopsis* sp. and *D. nuchalis*, there were 40 bp and 16 bp between *trnY* and *COX1*, *COX2* and *trnK*, respectively. For *D. ineffectus* and *R. stellata*, the two identical intergenic spacers were 2 bp between the *ND4L-trnT* and *trnP*-*ND6* junction, respectively (Appendix A).

### 3.4. Transfer and Ribosomal RNA Genes

The positions of all 22 typical transfer RNA genes (tRNAs) scattered throughout the whole mitogenomes were located in *D*. *ineffectus* and *R*. *stellata* (Appendix A). Among them, 14 tRNAs are encoded on the J-strand and the remaining eight on the N-strand. The total length of the 22 tRNAs was 1439 bp in *D. ineffectus* and 1441 bp in *R. stellata*, accounting for 9.1% and 9.4% of their whole genomes, respectively (Appendix A). The sizes of the 22 tRNAs range from 62 (*trnA*, *trnR*) to 70 bp (*trnK*) in *D. ineffectus* and from 61 (*trnA*) to 73 bp (*trnW*) in *R. stellata* (Appendix A). All 22 tRNAs in these two mitogenomes indicated a positive AT-skew (0.014, 0.009) and GC-skew (0.188, 0.15) (Appendix A).

All tRNAs could be folded into the canonical cloverleaf secondary structure including the aminoacyl (or acceptor) arm, dihydrouridine (DHU) arm, anticodon arm and pseudouridine (TΨC) arm except for the *trnS1*, which lacks the DHU arm and is replaced by a simple loop in these two new mitogenomes (Figure 3 and Figure 4), as is found in other deltocephaline leafhoppers [34,36]. The missing DHU arm concerning the *trnS1* probably appeared very early in the evolution of the Metazoa [52], and is frequent in insect mitogenomes [12]. Based on the predicted secondary structure, the size of the anticodon loop of all tRNAs is highly conserved with 7 bp compared with the variable length of DHU and TΨC loops (Figure 3 and Figure 4). In addition to the classic AU and GC pairs, a total number of 26 GU, 9 UU, 2 AA, 2 AC and 1 GA unmatched base pairs was found in *D. ineffectus*, while 23 GU, 9 UU, 1 AA and 1 AC mismatched base pairs was observed in *R. stellata* (Figure 3 and Figure 4). Moreover, in these two Drabescini mitogenomes, there was also an unpaired base (single A/C nucleotide) in the aminoacyl arm of *trnR* and the TΨC arm of *trnC*.

The two rRNA genes (*rrnL* and *rrnS*) are encoded on the N-strand in *D. ineffectus* and *R. stellata*. The large rRNA (*rrnL*), located between *trnL1* and *trnV*, ranges in length from 1210 bp (*R. stellata*) to 1220 bp (*D. ineffectus*), while the small rRNA (*rrnS*) located between *trnV* and the A + T-rich region ranges in size from 721 bp (*D. ineffectus*) to 743 bp (*R. stellata*) (Appendix A). These two rRNAs with a heavy AT nucleotide bias reach 80% in *D. ineffectus* and 80.4% in *R. stellata*, respectively (Appendix A); this is also found in other sequenced Drabescini species. Additionally, the two newly sequenced mitogenomes show the negative AT-skew (−0.120, −0.098) and positive GC-skew (0.247, 0.264) in rRNAs (Appendix A). Consequently, the rRNAs are highly conserved in the Cicadellidae.

### 3.5. A + T-Control Region

The putative A + T-control region of Drabescini mitogenomes is located between *rrnS* and *trnI*, ranging in size from 956 bp to 1381 bp except for partial mitogenomes (*Athysanopsis* sp. and *Dryadomorpha* sp.) (Appendix A). This region is deemed to be related to the origin of replication and transcription [11,12]. The control region of *D. ineffectus* is 1381 bp in length with an AT content of 85.3%, while *R. stellata* is 983 bp in length with an AT content of 74.7% (Appendix A). These tandem repeats in the control region have been reported in other sequenced deltocephaline mitogenomes [32], and also found in Drabescini species, indicating that these different fragment lengths and types of absolute tandem repeat regions are present in the two taxa. The A + T-control region of *D*. *nuchalis* has one kind of tandem repeat including two 167 bp repeat units and a partial sequence (84 bp) ranging in nucleotide positions from 13 bp to 418 bp. Two types of T/A tandem repeats are present in *D. ineffectus* with small size of 47 bp and 50 bp (Figure 5). However, no tandem repeat unit was found in *R. stellata*. As in the Drabescini mitogenomes, tandem repeat regions are common and the variable length and copy number of repeat units point to a conspicuous divergence of A + T-control region.

### 3.6. Nucleotide Diversity and Evolutionary Rate Analysis

The sliding window analysis concerning the nucleotide diversity (Pi values) of the 13 aligned PCGs among the five Drabescini mitogenomes *Athysanopsis* sp., *D*. *nuchalis*, *Dryadomorpha* sp., *D*. *ineffectus* and *R. stellata* are shown in Figure 6A. This exhibits the high degree of nucleotide variation within different genes. Nucleotide diversity values range from 0.176 (*COX1*) to 0.348 (*ATP8*) in these five species. In all PCGs, the *ATP8* (Pi = 0.348) presents the highest variability next to *ND6* (Pi = 0.308), *ND2* (Pi = 0.271) and *ND3* (Pi = 0.253) showing the comparatively high nucleotide diversity values. The *ND1* (Pi = 0.200), *ND4L* (Pi = 0.200), *COX3* (Pi = 0.192) and *COX1* (Pi = 0.176) with relatively low nucleotide diversity values indicate that they are relatively conserved genes in 13 PCGs (Figure 6A). To further analyze the evolutionary rate of PCGs, the ratio of Ka/Ks (ω) was used to estimate the selective pressure for each PCGs under positive selection, neutral evolution or purifying selection. As shown in Figure 6B, it is observed that all ratios of Ka/Ks (0 < ω < 1) are less than 1, indicating these PCGs are evolving under a purifying selection. Among the 13 PCGs, *COX1* (ω = 0.080) has undergone the strongest purifying selection and exhibits the lowest evolutionary rate. By contrast, *ATP8* (ω = 0.726) and *ND6* (ω = 0.442) have undergone comparatively weak purifying pressure, demonstrating a relatively fast evolutionary rate. Furthermore, pairwise genetic distances among these five mitogenomes also yield similar results. The average values show that *ATP8* (0.472), *ND6* (0.399) and *ND2* (0.338) with a high distance are evolving comparatively fast, while *ND4L* (0.233), *COX3* (0.223) and *COX1* (0.201) with a low distance are evolving relatively slow.

In this case, nucleotide diversity analyses in terms of other gene regions are significant for further identifying potential markers in future studies focusing in Drabescini species [53]. The *COX1* presents low variation and the lowest evolution among PCGs, and it is regarded as the universal barcode for species identification and delimitation [54], particularly in Deltocephalinae with close and ambiguous morphological characters.

### 3.7. Phylogenetic Relationships

The Bayesian inference (BI) and maximum likelihood (ML) phylogenetic analyses among Drabescini species was conducted based on three different datasets (PCG123, PCG12R and AA). These results indicated that the phylogenetic topologies are consistent, with most branches receiving strong support (Figure 7 and Appendix A). Our putative ingroup was recovered as monophyletic with respect to Cicadoidea and Cercopoidea. The inferred relationships based on the PCG123 and AA datasets Iassinae + Coelidiinae are sister to a clade comprising Megophthalminae and treehoppers with moderate to high values, which is consistent with previous phylogenetic analyses [34,42,43]. For the PCG12R datasets, Iassinae + Coelidiinae and Deltocephalinae are grouped into a clade. Another four cicadellid subfamilies Cicadellinae, Eurymelinae, Ledrinae and Typhlocybinae as currently recognized were recovered here as monophyletic with strong branch support. However, the relationships among the major lineages in the subfamily Evacanthinae remain poorly resolved and are not recovered as monophyletic.

These results provide a well-resolved phylogenetic topology for Deltocephalinae with moderate to high support for most branches. In agreement with previous studies [1,8,9,10], our phylogenetic analyses based on three different datasets using both BI and ML methods showed a strong support for the monophyly of Deltocephalinae. The present analyses consistently recovered Macrostelini as sister to the remaining tribes of this subfamily (PP = 100%, BS = 100), as have previous phylogenetic studies [28,34]. Additionally, Drabescini was recovered as monophyletic with high posterior probabilities and bootstrap support value in BI and ML trees (PP = 100%, BS = 100), as were Chiasmini, Cicadulini, Deltocephalini and Scaphoideini. In particular, the sister group of Drabescini is a clade comprised of the five included representatives of Scaphoideini with strong support, also suggested by recent phylogenetic analyses based on combined morphological and molecular data that confirm that Drabescini is closely related to Scaphoideini [1]. Nevertheless, the remaining three tribes including Athysanini, Opsiini and Paralimnini were found not to be a monophyletic group in our analyses.

Within the Drabescini, five species representing two subtribes Drabescina and Paraboloponina which were previously placed in a separate subfamily Selenocephalinae and treated as two tribes [2,3], form a monophyletic group with high branch support. Our analyses provided distinct evidence of close relationships between Deltocephalinae and Selenocephalinae, suggesting that the latter was not distinguishable from Deltocephalinae. While these results indicate consistent support for the monophyly of this tribe, its internal topologies diverge when using different datasets. Phylogenetic analyses among five Drabescini species based on PCG123-BI and PCG123-ML methods showed the relationships (*R*. *stellata* + (*Dryadomorpha* sp. + (*D*. *nuchalis* + (*Athysanopsis* sp. + *D. ineffectus*)))). On the other hand, the PCG12R-BI and PCG12R-ML analyses yielded the topologies (*R. stellata* + (*D. ineffectus* + (*Athysanopsis* sp. + *D. nuchalis*))) and (*R. stellata* + (*D. nuchalis* + (*Athysanopsis* sp. + *D. ineffectus*))), respectively. Moreover, the inferred relationships based on amino acid sequences (*R. stellata* + ((*Athysanopsis* sp. + *D. nuchalis*) + (*D. ineffectus* + *Dryadomorpha* sp.))) were also recovered with high support values. However, these studies were not retrieved as congruent results based on current mitogenome data. Therefore, further samples should be added to elucidate the status and relationships and improve the resolution of the still poorly-supported and varied branches among the major lineages within Membracoidea.

## 4. Conclusions

In this study, we determined two newly complete mitogenomes including *D*. *ineffectus* and *R*. *stellata* and found them consistent with previously reported mitogenomes of Cicadellidae. The Drabescini mitogenomes are highly conserved in base content and composition, genome size and order, protein-coding genes and codon usage, and secondary structure of tRNAs. The BI and ML phylogenetic analyses among the major lineages based on the concatenated datasets (PCG123, PCG12R and AA) yielded the well-resolved topologies with moderate to high support for most branches except for a few deep internal nodes within Membracoidea. While the relationships among tribes remain poorly resolved within Deltocephalinae, Drabescini was recovered as monophyletic with strong branch support and revealed close relationships (*R*. *stellata* + (*Dryadomorpha* sp. + (*D*. *nuchalis* + (*Athysanopsis* sp. + *D*. *ineffectus*)))). Furthermore, these results provide the comprehensive framework and valuable data toward the future resolution of phylogenetic relationships in this tribe.

## Figures and Tables

**Figure 1 insects-11-00534-f001:**
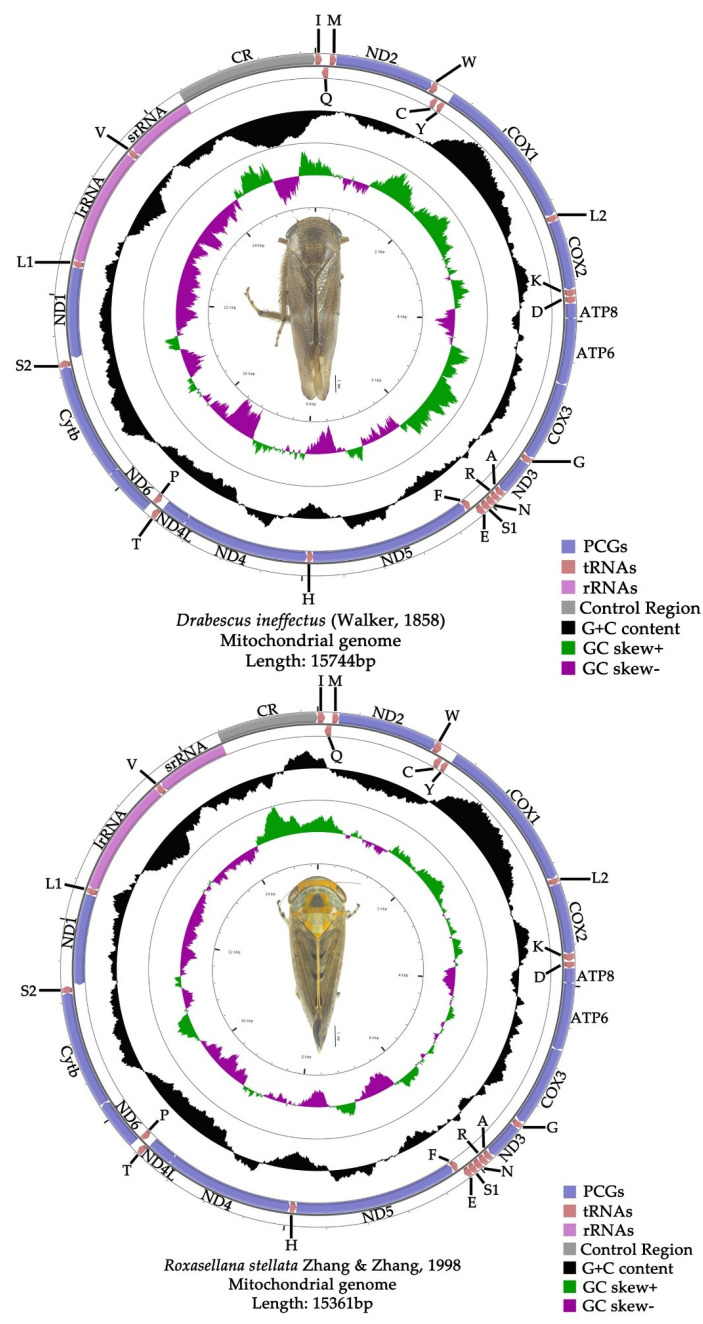
Circular map of the mitochondrial genome of *Drabescus ineffectus* and *Roxasellana stellata.*

**Figure 2 insects-11-00534-f002:**
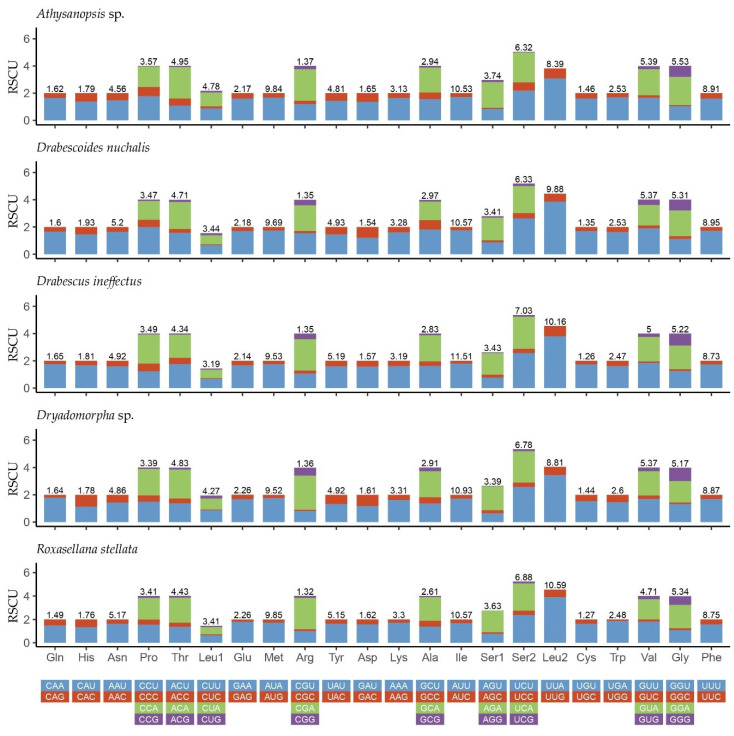
Relative synonymous codon usage (RSCU) in the mitogenomes of five Drabescini species.

**Figure 3 insects-11-00534-f003:**
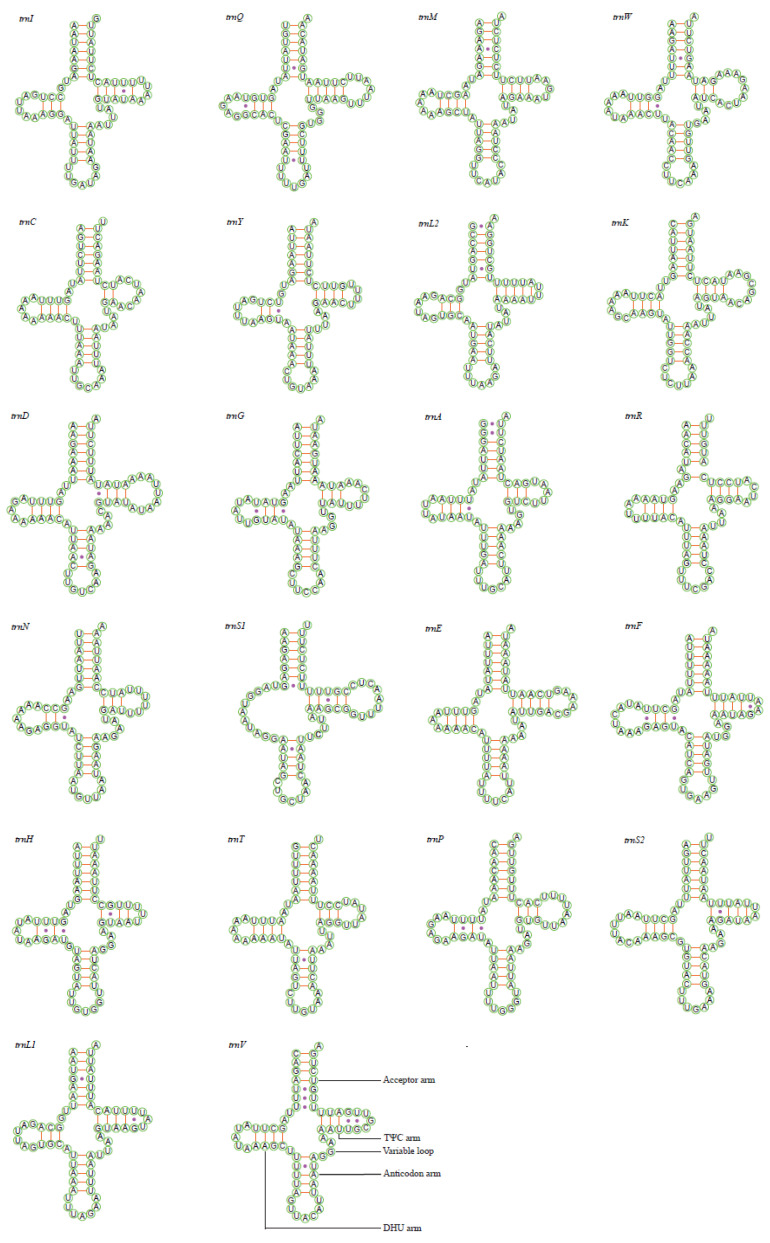
Predicted cloverleaf secondary structure for the 22 tRNAs of *Drabescus ineffectus*.

**Figure 4 insects-11-00534-f004:**
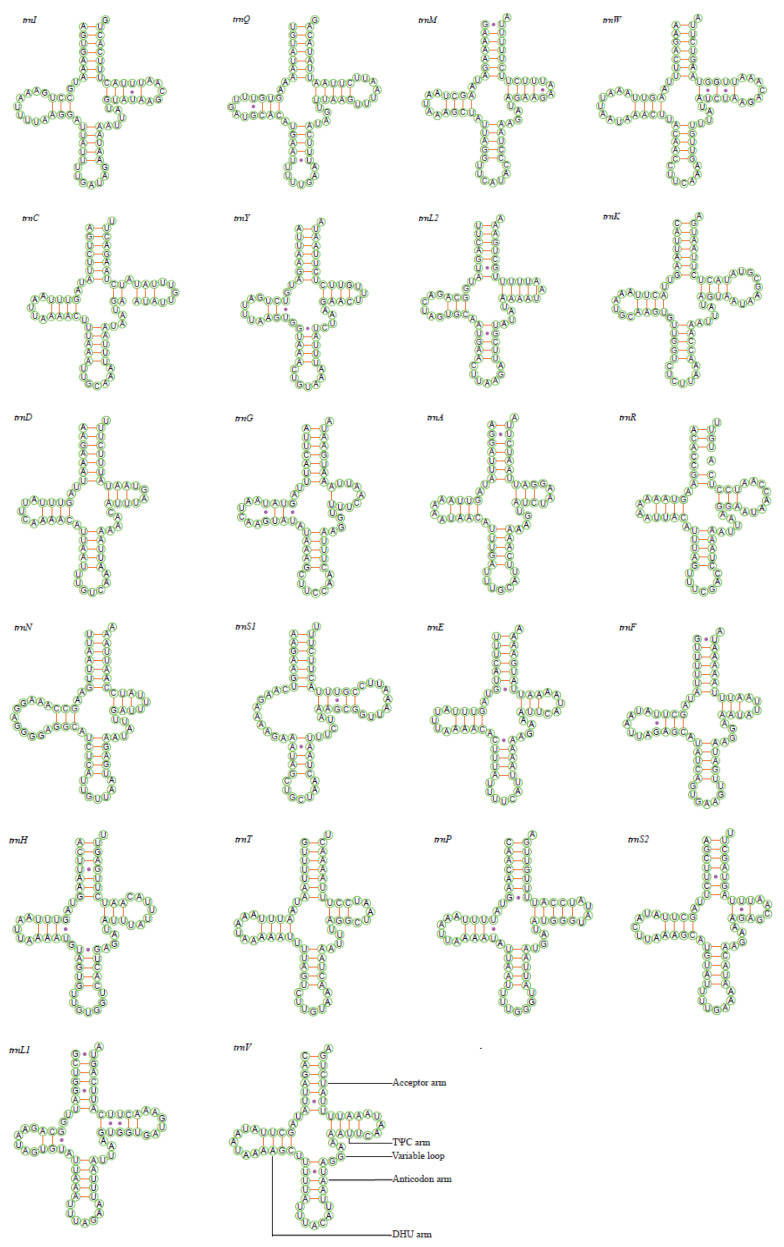
Predicted cloverleaf secondary structure for the 22 tRNAs of *Roxasellana stellata*.

**Figure 5 insects-11-00534-f005:**
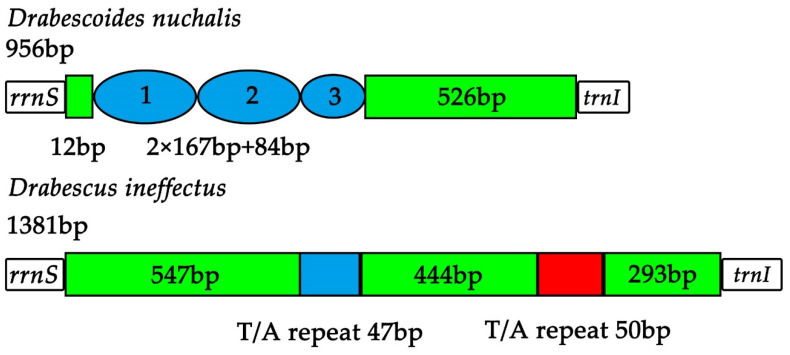
Structures of the A + T-control region in Drabescini mitochondrial genomes. The blue ovals indicate the tandem repeats. The blue and red blocks represent the T/A repeat regions.

**Figure 6 insects-11-00534-f006:**
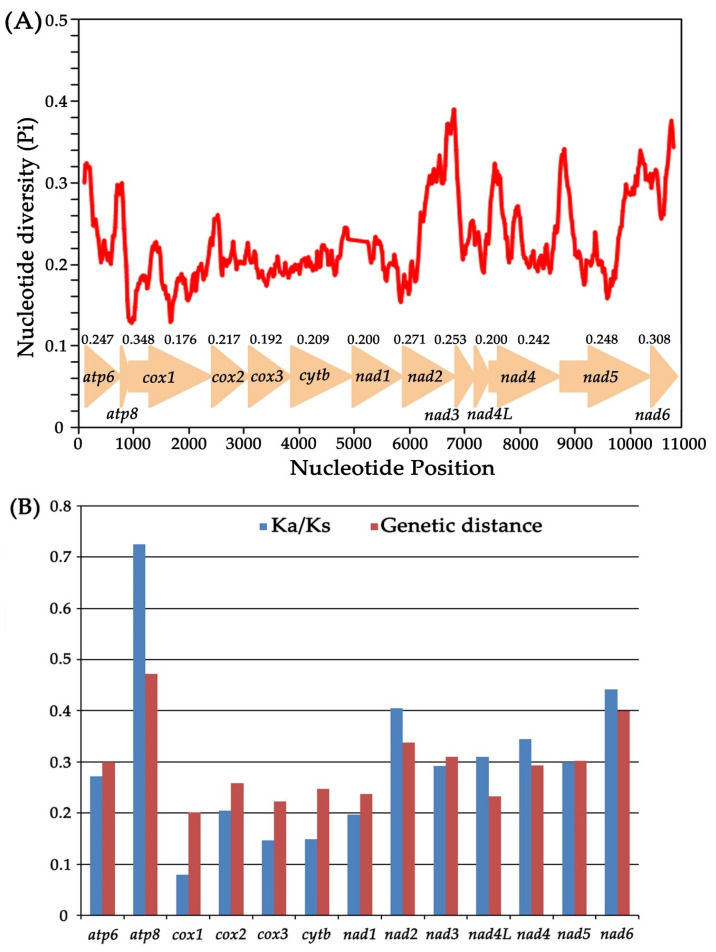
(**A**) Sliding window analysis of 13 aligned PCGs among five Drabescini mitogenomes. The red curve shows the value of nucleotide diversity (Pi). (**B**) Ratio of non-synonymous (Ka) to synonymous (Ks) substitution rates and genetic distance (on average) of 13 PCGs among five Drabescini mitogenomes.

**Figure 7 insects-11-00534-f007:**
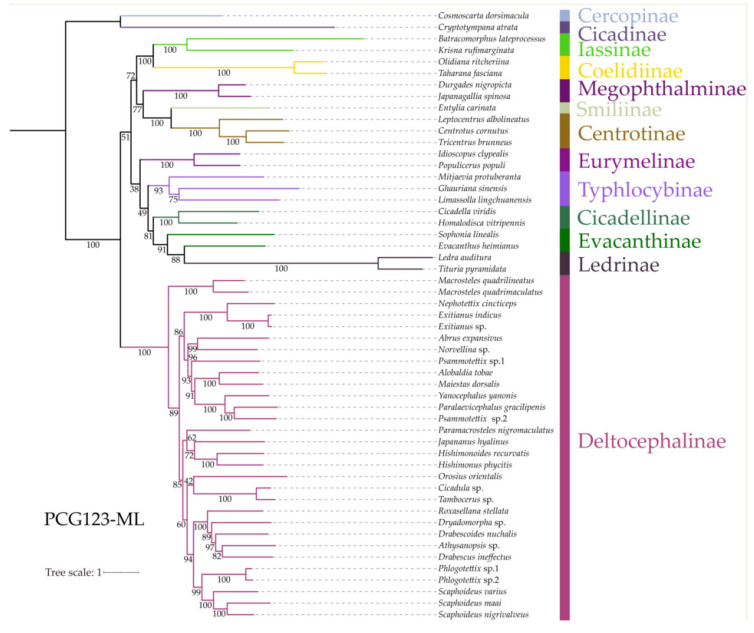
Phylogenetic tree inferred from ML method based on PCG123 dataset. Numbers on branches are bootstrap support values (BS).

**Table 1 insects-11-00534-t001:** Mitochondrial genomes used for phylogenetic analysis in present study.

Superfamily	Family	Subfamily	Species	Accession Number	Reference
Cicadoidea	Cicadidae	Cicadinae	*Cryptotympana atrata*	JQ910980	[25]
Cercopoidea	Cercopidae	Cercopinae	*Cosmoscarta dorsimacula*	NC_040115	[26]
Membracoidea	Membracidae	Smiliinae	*Entylia carinata*	NC_033539	[27]
		Centrotinae	*Centrotus cornutus*	KX437728	[16]
			*Leptocentrus albolineatus*	MK746137	[28]
			*Tricentrus brunneus*	NC_044708	[28]
	Cicadellidae	Deltocephalinae	*Abrus expansivus*	NC_045238	[29]
			*Norvellina* sp.	KY039131	[30]
			*Paramacrosteles nigromaculatus*	NC_045270	Direct Submission
			*Tambocerus* sp.	KT827824	[31]
			*Exitianus* sp.	KX437722	[16]
			*Exitianus indicus*	KY039128	[30]
			*Nephotettix cincticeps*	NC_026977	Direct Submission
			*Cicadula* sp.	KX437724	[16]
			*Alobaldia tobae*	KY039116	[30]
			*Maiestas dorsalis*	NC_036296	[32]
			*Athysanopsis* sp.	KX437726	[16]
			*Drabescoides nuchalis*	NC_028154	[17]
			*Drabescus ineffectus*	MT527188	This study
			*Roxasellana stellata*	MT527187	This study
			*Dryadomorpha* sp.	KX437736	[16]
			*Macrosteles quadrilineatus*	KY645960	[33]
			*Macrosteles quadrimaculatus*	NC_039560	[34]
			*Hishimonus phycitis*	KX437727	[16]
			*Hishimonoides recurvatis*	KY364883	Unpublished
			*Japananus hyalinus*	NC_036298	[32]
			*Orosius orientalis*	KY039146	[30]
			*Paralaevicephalus gracilipenis*	MK450366	[35]
			*Psammotettix* sp.1	KX437725	[16]
			*Psammotettix* sp.2	KX437742	[16]
			*Yanocephalus yanonis*	NC_036131	[30]
			*Phlogotettix* sp.1	KY039135	[30]
			*Phlogotettix* sp.2	KX437721	[16]
			*Scaphoideus maai*	KY817243	[36]
			*Scaphoideus nigrivalveus*	KY817244	[36]
			*Scaphoideus varius*	KY817245	[36]
		Cicadellinae	*Cicadella viridis*	MK335936	[37]
			*Homalodisca vitripennis*	NC_006899	Direct Submission
		Coelidiinae	*Olidiana ritcheriina*	NC_045207	Direct Submission
			*Taharana fasciana*	KY886913	[38]
		Eurymelinae	*Idioscopus clypealis*	NC_039642	[39]
			*Populicerus populi*	MH492318	[40]
		Evacanthinae	*Evacanthus heimianus*	MG813486	[41]
			*Sophonia linealis*	KX437723	[16]
		Iassinae	*Batracomorphus lateprocessus*	MG813489	[42]
			*Krisna rufimarginata*	NC_046068	[42]
		Ledrinae	*Ledra auditura*	MK387845	[43]
			*Tituria pyramidata*	NC_046701	Direct Submission
		Megophthalminae	*Durgades nigropicta*	KY123686	[44]
			*Japanagallia spinosa*	NC_035685	[44]
		Typhlocybinae	*Mitjaevia protuberanta*	NC_047465	Unpublished
			*Ghauriana sinensis*	MN699874	[45]
			*Limassolla lingchuanensis*	NC_046037	Unpublished

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
