# Peer review of "Characterization of the Complete Mitochondrial Genome of *Drabescus ineffectus* and *Roxasellana stellata* (Hemiptera: Cicadellidae: Deltocephalinae: Drabescini) and Their Phylogenetic Implications"

_insects, 2020, doi:10.3390/insects11080534_

Round 1
Reviewer 1 Report
Nicely written and presented contribution to the mitogenomics of the very large and complex subfamily of Cicadellidae - next step to the knowledge of the group based on two more species of Drabescini. Current classification and relationships within Cicadellidae are complex issues and all new data are welcome. Some minor updates are necessary, marked on attached file. Main text and supplementary files are well presented.
The paper presents for the first time two new complete mitogenomes of the Drabescini Deltocephalinae leafhoppers (Cicadellidae) species. It is another contribution to mitogenomic sequences of this largest and most differentiated family.
The results presented are new data for Drabescini, for which tribe only a mitogenomes for a very few other taxa were known. This is important as this tribe is now split into two subtribes, so more data are necessary to cover the diversity of the group. Drabescini is one of the tribes of Deltocephalinae - the most diverse and speciose group of Cicadellidae, with numerous taxonomic and classification questions to be solved. Therefore the mitogenomic data could be used and important set for phylogenetic and evolutionary studies.

Author Response
Response to Reviewer 1 Comments:
Point 1: Line-32. zoo-geographic
Response 1: Accepted and changed to “zoogeographic”.
Point 2: Line-36. lights
Response 2: Accepted and changed to “light”.
Point 3: Line-117(PDF). Cryptotympana atrata
Response 3: Accepted. The species names are italicized.
Point 4: Line-117(PDF). Cosmoscarta dorsimacula
Response 4: Accepted. The species names are italicized.
Reviewer 2 Report
I think this is an excellent, and very thorough study. In fact, I think that it could be broken into two studies with the mitogenomes as one manuscript and the phylogenetic analysis as a second manuscript. If the paper was broken into two studies, more detail could be given to the phylogenetic portion allowing for more detail in the figures and explanation of the phylogenies. In its present form, some minor notes about language should help the paper.
Line 47 - change word controversial. All data up to this sentence suggests a monophyletic tribe so the previous studies are not controversial but the lack of DNA evidence suggests that more data is needed to obtain better information about the tribe. So I would suggest changing that word.
Line 61-67. This sentences is too long and should be broken up into two sentences for clarity.
Line 89-92 - This sentence starting with "All 13 PCGs..." is a bit confusing and should be reworded for clarity
Line 114 - Use of the word "performed" is incorrect. It could be substituted with collected or maybe assembled. Or removed altogether to say "A total of 53 mitogenomes of Cicadomorpha insects were phylogenetically analyzed."
Line 117 - two insect names are not italicized, nor are the authors cited.
Line 205-207: I had difficulty understanding this sentence, but I am not sure it can be improved. Sometimes sharing results is just difficult.
Line 213-215: same comment as above, it is difficult to follow this sentence
Line 216: it is not definitively clear which species is 20bp and which is 17bp.
Line 285 - This is just personal but when I think of things under purifying selection, I would not say they are evolving more quickly than other genes. Maybe just that the genes with the very lowest Ka/Ks are under the strongest selection to remain intact but all other genes don't seem to have the same pressure. But this is really just a personal interpretation so it doesn't need to be changed if the authors wish to interpret the results the way they have.
Line 310-338 - This whole section was difficult for me to follow as I work on Hemiptera, but not specifically on this group. I think one of three things should happen: 1) It should be extracted from the paper and made into its own work so the authors can have more detailed images with more labels and more detail explanation. I think the amount of work they have down warrants a separate paper and it would allow for much more detail and it is clear the authors understand the system enough to include that detail, 2) some of the deeper groups they cite should be labeled on one of the figures so it is easier to go back and forth between the text and find the groups on the figure, or 3) scale it down a lot in the paper because it is hard to follow without a figure showing the information explicitly.
Other than those minor comments I think the work is VERY commendable and worthy of publication. They clearly know the system and the did a large amount of work for this manuscript.
